# EBF1 Negatively Regulates Brassinosteroid-Induced Apical Hook Development and Cell Elongation through Promoting BZR1 Degradation

**DOI:** 10.3390/ijms232415889

**Published:** 2022-12-14

**Authors:** Na Zhao, Min Zhao, Lingyan Wang, Chao Han, Mingyi Bai, Min Fan

**Affiliations:** The Key Laboratory of Plant Development and Environmental Adaptation Biology, Ministry of Education, School of Life Science, Shandong University, Qingdao 266237, China

**Keywords:** brassinosteroid, EBF1, BZR1, EIN3, PIF4

## Abstract

Brassinosteroids (BRs) are a group of plant steroid hormones that play important roles in a wide range of developmental and physiological processes in plants. Transcription factors BRASSINOZALE-RESISTANT1 (BZR1) and its homologs are key components of BR signaling and integrate a wide range of internal and environmental signals to coordinate plant growth and development. Although several E3 ligases have been reported to regulate the stability of BZR1, the molecular mechanism of BZR1 degradation remains unclear. Here, we reveal how a newly identified molecular mechanism underlying EBF1 directly regulates BZR1 protein stability via the 26S proteasome pathway, repressing BR function on regulating Arabidopsis apical hook development and hypocotyl elongation. BZR1 directly binds to the EBF1 gene promotor to reduce EBF1 expression. Furthermore, the genetic analysis shows that BZR1, EIN3 and PIF4 interdependently regulate plant apical hook development. Taken together, our data demonstrates that EBF1 is a negative regulator of the BR signaling pathway.

## 1. Introduction

Brassinosteroids (BRs), a group of plant steroid hormones, play critical roles in the regulation of a wide range of plant growth and development processes, such as cell elongation, seeds development processes, stomatal closure, stress tolerance, skotomorphogenesis and photomorphogenesis [1,2,3]. BRs are recognized by the cell surface receptor kinase BRASSINOSTEROID INSENSITIVE1 (BRI1), triggering a series of downstream phosphorylation and dephosphorylation, and finally activating the core transcription factor BRASSINAZOLE-RESISTANT1 (BZR1) and its homologues. In the case of low levels of BR, BZR1 is phosphorylated by phosphokinase BRASSINOSTEROID-INSENSITIVE 2 (BIN2), and it is exported from the nucleus, and then, it degrades [2,4].

BZR1 and BES1 (BRI1-EMS-SUPPRESSOR1) represent a class of plant-specific transcription factors, and they interact with other transcript factors, such as PIF4 (PHYTOCHROME INTERACTING FACTOR4), ARF6 (AUXIN RESPONSE FACTOR6) and EIN3 (ETHYLENE-INSENSITIVE 3) to integrate BR; they also interact with other hormones and environmental signals to regulate various plant physiological processes [5,6,7]. The activities of BZR1 and BES1 are modulated by multiple modes of regulation. Phosphorylation by BIN2 inactivates BZR1 and BES1, whereas dephosphorylation by PP2A (PROTEIN PHOSPHATASE 2A) activates BZR1 and BES1. H_2_O_2_-induced oxidation increases the transcription activity of BZR1 and BES1 by enhancing the interaction between BZR1 and BES1 with PIF4 and ARF6 [8]. Several families of E3 ubiquitin ligase have been reported to target BZR1 or BES1 and promote their degradation in response to different environmental signals. PUB40 (PLANT U-BOX40) is reported to mediate the degradation of BZR1 in Arabidopsis roots [9]. COP1 (CONSTITUTIVE PHOTOMORPHOGENIC1) mediates the degradation of phosphorylated BZR1 in darkness, and SINAT (SINA of *Arabidopsis thaliana*) mediates dephosphorylated BZR1 degradation under light [10,11]. The negative regulation factor of autophagy, TOR (TARGET OF RAPAMYCIN) kinase, could stabilize the BZR1 protein level, indicating that BZR1 could be degraded by the autophagy pathway [12].

EIN3 BINDING F-BOX PROTEIN1 (EBF1), an F-box protein containing 16 tandem leucine-rich repeats (LRRs), is a nuclear-localized E3 ubiquitin ligase. The EBF1 gene is expressed in all plant organs, indicating the vital roles played by EBF1 in plant development processes [13,14]. The protein structure of EBF1 is related to the budding yeast GRR1 protein, which is involved in cell cycle control and glucose signaling [15]. EBF1 and its paralog EBF2 were first described as negative regulators of the ethylene signaling pathway. EBF1/2 interact with the key components of the ethylene signaling pathway ETHYLENE-INSENSITIVE3 (EIN3)/EIN3-Like (EIL1), triggering the ubiquitination degradation of EIN3/EIL1 [16]. Plants overexpressing EBF1/2 exhibited insensitivity to ethylene [14]. In addition to the regulation of EIN3/EIL1 protein stability, EBF1 also mediates the degradation of PHYTOCHROME-INTERACTING-FACTOR3 (PIF3), which is a central regulator of the light signaling pathway [17]. EBF1/2 interact with PIF3 and promote the light-induced ubiquitination and degradation of PIF3 to regulate plant photomorphogenesis [18]. EIN3/EIL1 interact with PIF3 to form a transcriptional complex and regulate the expression of hundreds of common target gene expression, thereby controlling a wide range of plant growth and development processes.

In this study, we further demonstrated that EBF1 inhibits BR-induced hypocotyl elongation and apical hook development. BZR1 and BES1 interact with EBF1/2. EBF1 regulates the stability of the BZR1 protein via the 26S proteasome pathway. BZR1 associates with the EBF1 promoter and inhibits the expression of the EBF1 gene. In addition, BZR1, EIN3 and PIF4 interdependently promote the development of the apical hook. Taken together, the findings of our study indicate that a molecular mechanism underlying E3 ubiquitin ligase EBF1 negatively regulates the BR signaling pathway.

## 2. Results

### 2.1. EBF1 Inhibits BR-Induced Apical Hook Development and Hypocotyl Elongation

Our previous results showed that BR and ethylene promote apical hook development and hypocotyl elongation through the direct interaction between BZR1 and EIN3 [7]. To further determine the crosstalk of BR and ethylene in plant development, we analyzed how the key components of the ethylene signaling pathway respond to BR. In agreement with our previous results, a BR treatment was shown to promote apical hook formation and hypocotyl elongation, but these promoting effects of BR were significantly reduced in the *ein3 eil1* double mutant. The F-box proteins EBF1 and EBF2 negatively regulate ethylene signal transduction by promoting EIN3 and EIL1 degradation. The overexpression of EBF1 also displayed reduced BR sensitivity in both apical hook development and hypocotyl elongation. However, the lowered sensitivity of *ein3 eil1* to BR was partially suppressed by *ebf1 ebf2 ein3* (Figure 1A–D). For hypocotyl elongation, in the presence of low concentrations of BL, the hypocotyl length of the *ebf1 ebf2 ein3* triple mutant was similar to that of the *ein3 eil1* double mutant, whereas in the presence of high concentrations of BL, the *ebf1 ein3 eil1* triple mutant displayed the significant longer hypocotyl than the *ein3 eil1* double mutant did, suggesting that the mutation of EBF1 and EBF2 compensates the dwarf phenotypes of *ein3 eil1* under high levels of BR conditions. Consistent with this, the *ebf1 ebf2 ein3* triple mutant showed the more hypersensitive to BR compared to the *ein3 eil1* double mutant in etiolated apical hook development. Interestingly, the loss of functions in EBF1 and EBF2 repressed the apical defective growth phenotype of the *ein3 eil1* mutant even in the presence of low-dose BR. Together, these results suggested that EBF1 may be involved in the BR signaling pathway by regulating the stability of the proteins other than EIN3/EIL1.

### 2.2. EBF1 Interacts with BZR1 In Vivo and In Vitro

To determine how EBF1 inhibits the BR signaling pathway, we analyzed the interaction between EBF1/EBF2 and the positive regulators of the BR signaling pathway using the yeast two-hybrid assay. We found that BZR1 interacted with EBF1 in yeast (Figure 2A). BES1 also displayed an interaction with EBF1 and EBF2 in yeast (Appendix A). The protein pull-down assays showed that the glutathione S-transferase (GST) fusion protein GST-BZR1 interacted with the maltose binding protein (MBP) fusion protein MBP-EBF1, but it did not interact with MBP (Figure 2B). To test whether EBF1 interacts with BZR1 in plants, we performed the BiFC (bimolecular fluorescence complementation) assay in Arabidopsis mesophyll protoplast cells. The results showed that BZR1-cYFP co-transformed with EBF1-nYFP in Arabidopsis mesophyll protoplast cells displayed the strong yellow fluorescence signal, indicating that BZR1 interacted with EBF1 in the plant (Figure 2C). Consistent with BiFC, the interaction of EBF1 and BZR1 in the plants was confirmed using a co-immunoprecipitation (Co-IP) assay that expresses *pro35S:BZR1-GFP* and *pro35S:EBF1-MYC* in protoplasts (Figure 2D). These results indicated that BZR1 interacts with EBF1 in vivo and in vitro.

### 2.3. EBF1 Inhibits the Expression of BZR1 Target Genes

To examine the effects of EBF1 on BZR1 functions, we analyzed the expression of BZR1 target genes in wild-type and *pro35S:EBF1-Myc* transgenic plants. *EXP8*, *SAUR15* and *PRE5* have been reported as BZR1 target genes which are related to cell elongation [6,19]. The results showed that the BR treatment significantly increased the transcript levels of *EXP8* and *SAUR15*, but these promoting effects of BR were reduced in *pro35S:EBF1-Myc* transgenic plants (Figure 3A). To further verify the inhibiting effects of EBF1 on BZR1 activity, we performed the transient expression assay using the Arabidopsis mesophyll protoplast. We observed that the luciferase activity derived from *proPRE5:LUC* significantly increased when BZR1 was transfected alone, while co-transfected BZR1 and EBF1 or EBF2 significantly reduced the transcriptional activity of BZR1 (Figure 3B). These results indicated that EBF1 inhibits the activity of BZR1.

### 2.4. EBF1 Promotes the Degradation of BZR1

Considering that EBF1/EBF2 inhibit BZR1 activity and EBF1/EBF2 are E3 ligases, we speculated that EBF1 may be involved in BZR1 degradation. To test this hypothesis, we examined the protein levels of BZR1 in 10-day-old Arabidopsis seedlings of *pro35S:BZR1-GFP* and *pro35S:BZR1-GFP/pro35S:EBF1-MYC* transgenic plants. The quantitative RT-PCR showed that the EBF1 expression levels are similar in *pro35S:EBF1-GFP* and *pro35S:BZR1-GFP/pro35S:EBF1-MYC* transgenic plants, and that the *BZR1* expression levels are similar in *pro35S:BZR1-GFP* and *pro35S:BZR1-GFP/pro35S:EBF1-MYC* transgenic plants (Appendix A). However, the BZR1 protein level in the *pro35S:BZR1-GFP/pro35S:EBF1-MYC* was significantly lower than it was in the *pro35S:BZR1-GFP* plant (Figure 4A,B). Consistent with this, the fluorescent signals of BZR1-GFP in *pro35S:BZR1-GFP* and *pro35S:BZR1-GFP/pro35S:EBF1-MYC* transgenic plants hypocotyl cells also showed that the nuclear-localized BZR1 level in *pro35S:BZR1-GFP* was higher than that it was in the *pro35S:BZR1-GFP/pro35S:EBF1-MYC* plant (Figure 4C,D). The stability of BZR1 was reduced after the seedlings were transferred to darkness from continuous light conditions, and this reduction was enhanced in the EBF1 overexpression plants (Appendix A). Further, the proteasome inhibitor MG132 efficiently inhibited BZR1 degradation in the background of both the wild-type and EBF1 overexpression transgenic plants (Figure 4E). These results indicate that EBF1 promotes BZR1 degradation by the 26S proteasome pathway.

### 2.5. BZR1 Reduces EBF1 Expression

The published BZR1 ChIP-Seq results show that BZR1 exhibited a binding peak on the promoter of EBF1 [6]. To examine whether EBF1 is a target of BZR1, we performed a ChIP-qPCR using the *pro35S:BZR1-YFP* and *pro35S:YFP* transgenic plants (Figure 5A). There are two G-box elements (CACGTG) and one E-box element (CATGTG) in the EBF1 promoter. The results showed that BZR1 highly enriched in the promoter of EBF1. To investigate whether BZR1 regulates the expression of EBF1, we analyzed the expression of EBF1 in the mesophyll protoplast. The results showed that BZR1 significantly reduced the luciferase activity that is derived from *proEBF1* (Figure 5B). These results indicated that the BZR1 associates with the EBF1 promoter to inhibit its expression.

### 2.6. BZR1 Needs at Least One Class of Transcription Factor EIN3 or PIFs to Promote Apical Hook Development

Previous studies showed that EBFs promote PHYTOCHROME INTERACTING FACTOR3 (PIF3) degradation to regulate cell elongation and cold tolerance [18,20]. PIF4, the homolog of PIF3, has been reported to interact with BZR1 and EIN3 to regulate cell elongation. We therefore speculated that BZR1, EIN3 and PIF4 may form a transcription complex to regulate apical hook formation. To test this hypothesis, we analyzed the BR response in the *ein3 eil1*, *pifq* (*pif1 pif3 pif4 pif5*) and *pifq ein3 eil1* mutants. The BL treatment gradually increased the apical hook curvature of the *pifq* and *ein3 eil1* mutants in a dose-dependent manner. Meanwhile, the BL treatment did not lead to any change in the *pifq ein3 eil1* hook curvature (Figure 6A). HOOKLESS (HLS1) is a central regulator of Arabidopsis apical hook development [21]. Previous studies showed that BZR1 promotes *HLS1* expression [7], the transient assay result showed that BZR1 could promote *HLS1* expression in the Col-0, *ein3 eil1* and *pifq* seedlings, but it could the not in *pifq ein3 eil1* seedlings (Figure 6B). We also found that the gain-of-function mutant *bzr1-1D* partially restored the hookless phenotype of the *pifq* and *ein3 eil1* mutants, but it failed to promote hook development in the *pifq ein3eil1* mutant (Figure 6C,D). These results suggest that the promotion of hook development by BR and BZR1 requires at least one class of transcription factors PIFs or EIN3/EIL1.

## 3. Discussion

The assessment of protein stability is critical for the growth and survival of living organisms. The organism removes unwanted proteins via the ubiquitination pathway or the autophagy pathway. The ubiquitination pathway is a widespread mechanism in all eukaryotes. Many hormones signaling pathways are regulated by the SCF complex. For example, the SCF^SLY1^ complex mediates the degradation of the DELLAs-triggering gibberellin signaling pathway [22,23]. This paper illustrated that F-box protein EBF1 promotes BZR1 protein degradation by the 26S proteasome pathway, inhibiting the expression of BZR1 target genes and BR signaling. Meanwhile, BZR1 associates with the promoter of EBF1 and inhibits EBF1 gene expression, inducing BZR1 and EIN3 protein accumulation. Released BZR1 interacts with EIN3 and PIF4, regulating the target genes expression and plant growth.

BR activates the transcription factors BZR1 and its homologs, regulating a wide range of developmental and physiological processes in plants, such as seed development, stomatal development, xylem differentiation, root development, flowering, photomorphogenesis, skotomorphogenesis and so on [2,3]. BR signaling is intrinsically connected to other hormones and environmental signaling pathways through protein interaction and modification or transcription regulation between the key components of these signaling pathways. The interaction of BZR1 with PIF4, ARF6 and EIN3 mediates the integration of the BR, light, auxin and ethylene signaling pathways [5,6,7]. The negative regulator of gibberellin signaling pathway DELLAs inhibits the DNA binding activity of BZR1, restraining the BR signal transduction [19]. The BR signaling pathway is reported to be regulated by the ubiquitin pathway. COP1 promotes the degradation of phosphorylated BZR1 in darkness, and SINAT mediates the degradation of the dephosphorylated BZR1 under light [10,11]. PUB40 is reported to mediate the degradation of BZR1 in Arabidopsis roots [9].

In the present paper, our data demonstrated a new E3 ubiquitin ligase that regulates BZR1 protein degradation and BR signaling transduction. Firstly, in apical hook development and the hypocotyl elongation process, mutant *ebf1 ebf2 ein3* exhibited hypersensitivity to the BL treatment (Figure 1A–D). Secondly, EBF1 interacted with BZR1 and promoted BZR1 degradation via the 26S proteasome pathway. Thirdly, EBF1 inhibited the expression of the BZR1-induced genes, and finally, it negatively regulated BR signaling transduction (Figure 4A–E).

The expression of the EBF1 gene was also regulated by BZR1. Our study showed that BZR1 directly bonded to the EBF1 gene promoter and reduced the expression of the EBF1 gene (Figure 5A,B). Accumulated BZR1, EIN3 and PIF4 cooperatively promoted the development of the apical hook in Arabidopsis. BL promoted Arabidopsis seedlings’ apical hook development in Col-0, *ein3 eil1* and *pifq* mutants, but it did not do so in the *pifq ein3 eil1* mutants. BZR1 induced HLS1 gene expression and promoted apical hook development in the Col-0, *ein3 eil1* and *pifq* mutants, but it did not do so in the *pifq ein3 eil1* mutants (Figure 6A–D). These results indicate that to promote apical hook formation, BZR1 needs at least one of the transcription factors EIN3/EIL1 and PIF4. BZR1, EIN3/EIL1 and PIF4 form a transcription complex and synergistically regulate the expression of the target genes and plant growth. Recent research has shown that *SAUR17* (*SMALL AUXIN UP RNA17*) is another BZR1 target gene that is involved in regulating apical hook formation. BZR1, EIN3/EIL1 and PIF3/4 bind to the promoter of *SAUR17* and induce its expression to maintain the apical hook formation in the dark. The BZR1 induction of *SAUR17* expression is dependent on EIN3/EIL1 and PIFs. BZR1 promotes the protein stability of PIF3 and EIN3/EIL1 by inhibiting EBF1/2 expression [24]. This research partly supports our results.

One report shows that when the seedlings emerge from the soil, COP1, a negative regulator of light signals, interacts with EBF1/2, promoting the ubiquitination modification and protein degradation of EBF1/2 in the darkness at warm temperatures, inhibiting EIN3 degradation, and finally, promoting plant skotomorphogenesis [25]. COP1 was also reported to degrade the phosphorylated BZR1 in the dark [10]. In our study, the phosphorylated and dephosphorylated BZR1 protein levels were both regulated by EBF1 (Figure 4A–D). We speculate that COP1 mediates the degradation of EBF1 and phosphorylated BZR1 in the darkness, inducing the protein accumulation of dephosphorylated BZR1 and EIN3. BZR1, EIN3 and PIF4 cooperatively help the seedlings to carry out skotomorphogenesis, promoting the etiolated seedlings’ emergence from the soil. When the seedlings penetrate the soil, the EBF1 protein accumulates, leading to the degradation of BZR1 and EIN3, inhibiting the seedlings’ hypocotyl elongation under light and promoting the seedlings’ photomorphogenesis.

EBF1 was originally described as a negative regulator of the ethylene signaling pathway [16]. After years of research, more functions of EBF1 were discovered. In addition to being regulated by ethylene, EBF1 also respond to jasmonate at the transcription level in an apical hook formation process in seedlings [26]. EBF1 participates in plant photomorphogenic and freezing tolerance via mediating PIF3 protein degradation in response to light and temperature, respectively [18,20]. In this paper, we found that EBF1 also participates in the BR signaling pathway by regulating BZR1 protein stability. As the expression of EBF1 is extensive, more functions of EBF1 may be discovered in the future.

PIF3 plays a central role in plant growth, inhibiting seedling photomorphogenesis in the dark and regulating plant freezing tolerance by repressing the expression of *CBF* genes [17,20]. Previous studies showed that EBF1 interacts with PIF3 and decreases PIF3 protein stability via the 26S proteasome pathway, participating in the regulation of plant photomorphogenesis and cold stress response. PIF4 is a homologue of PIF3, which also regulates plant photomorphogenesis. In the present study, PIF4 collaborated with BZR1 and EIN3 to regulate plant apical hook development. We speculate that PIF4 may also be a target of EBF1, and it is degraded by EBF1. This hypothesis needs more evidence.

## 4. Materials and Methods

### 4.1. Plant Materials and Growth Conditions

All of the *Arabidopsis thaliana* seeds were grown on 1/2 Murashige and Skoog (MS) medium with or without the indicated treatment. The seeds were disinfected with 80% ethanol. After vernalization for 2 d at 4 °C, the seeds were grown in culture room under a 16 h-light/8 h-dark cycle at 22–24 °C. All of the mutants and transgenic materials were in Col-0 ecotype background. The *pifq* (*pif1 pif3 pif4 pif5*), *bzr1-1D pifq* and *pro35S:BZR1-GFP* were lab stocks. The *ein3 eil1*, *pifq ein3 eil1*, *ebf1 ebf2 ein3* and *pro35S:EBF1-MYC* were gifted from Hongwei Guo. *bzr1-1D ein3 eil1* were generated by *bzr1-1D* crossing with *ein3 eil1*. *bzr1-1D pifq ein3 eil1* were generated by *bzr1-1D pifq *crossing with *pifq ein3 eil1*.

### 4.2. Solution Preparation

Brassinolide (BL) and BR biosynthesis inhibitor propiconazole (PPZ) were purchased from Sigma-Aldrich (Louis, MO, USA). The BL was dissolved in 80% ethanol to prepare 2 mM stock solution. The PPZ was dissolved in DMSO to prepare 420 mM stock solution. For a mock treatment, ethanol or DMSO was diluted in 1/2 MS medium at the same fold dilution, respectively.

### 4.3. Hook Curvature Measurement

The seeds were grown on 1/2 MS medium with or without indicated treatment. After vernalization for 2 d at 4 °C, the seeds were grown in light for 6 h and then transferred to darkness for 4 days at 22–24 °C. Images of the individual hooks were acquired using a scanner, and the hook angles were measured using the ImageJ software (http://rsbweb.nih.gov/ij/, accessed on 1 February 2019). The angles were measured as described in [27].

### 4.4. Hypocotyl Length Measurement

The seeds were grown on 1/2 MS medium with or without the indicated treatment. After vernalization for 2 days at 4 °C, the seeds were grown in 16h-light/8h-dark circle growth chamber for 7 days at 22–24 °C. Images of the seedlings’ hypocotyl were acquired using a scanner, and the hypocotyl length was measured using the ImageJ software (http://rsbweb.nih.gov/ij/, accessed on 1 February 2019).

### 4.5. RNA Extraction, Reverse Transcription and Real-Time PCR

The total RNA was extracted from the seedlings which were grown on 1/2 MS medium for 7 days in in16h-light/8h-dark circle growth chamber using TransZol Up Plus RNA Kit (TransGen Biotech, Beijing, China). Then, the total RNA was subjected to reverse transcription at 42 °C for 1 h with Revert Aid reverse transcriptase (Thermo Fisher Scientific, Waltham, MA, USA). The quantitative PCR analyzes were performed on a CFX connect real-time PCR detection system (Bio-Rad, Hercules, CA, USA) using a SYBR green reagent (Roche, Basel, Switzerland) with gene-specific primers (Appendix A).

### 4.6. Yeast Two-Hybrid Assay

The full-length cDNA of BZR1, BES1 and BIN2 were cloned into the pAD-GAL4 vector as prey. The full-length cDNA of EBF1 and EBF2 were cloned into the pBD-GAL4 vector as the bait. The bait and every prey were co-transformed into the yeast strain AH109. The transformed yeast was grown on on medium SD (-Trp-Leu) and cultured, then screened for growth on medium SD (-Trp-Leu-His) with 1.5 mM 3-AT.

### 4.7. Pull-Down Assays

BZR1 fused to GST was purified from bacteria using glutathione beads (GE Healthcare, London, UK). EBF1 fused to MBP were purified using amylose resin (New England Biolabs, Ipswich, MA, USA). Glutathione beads containing 1 µg of GST-BZR1 were incubated with 1 µg MBP or MBP-EBF1 as indicated by the pull-down buffer (20 mM Tris-HCl pH7.5, 100 mM NaCl, 1 mM EDTA), and the beads were washed 10 times with the wash buffer (20 mM Tris-HCl pH 7.5, 300 mM NaCl, 0.5% TritonX-100, 1 mM EDTA). The eluted proteins were analyzed by immunoblot analysis with an anti-MBP antibody (New England Biolabs, Beijing, China, Cat: E8038L, 1:5000 dilution).

### 4.8. BiFC Assays

The full length of EBF1 or BZR1 were amplified by PCR and cloned into the pX-nYFP (*pro35S:X-nYFP*) or pX-cYFP (*pro35S:X-cYFP*) vector using the LR recombination reaction (Invitrogen, California, USA), respectively. The plasmids containing the BZR1-cYFP and pX-nYFP or EBF1-nYFP constructs were transiently transformed into the Arabidopsis protoplast. The protoplasts were kept in the greenhouse for 10 h at 22 °C. The fluorescent signals were visualized by using the LSM-880 laser scanning confocal microscope (Zeiss, Oberkochen, Germany).

### 4.9. Co-Immunoprecipitation Assays

The plasmids containing *pro35S:EBF1-MYC* with *pro35S:GFP* or *pro35S:BZR1-GFP* were transiently transform into the Arabidopsis Col-0 protoplasts. The protoplasts were kept in the greenhouse for 10 h at 22 °C. Then, the protoplasts were lysed with NEB buffer (20 mM HEPES-KOH, at pH 7.5, 40 mM KCl, 1 mM EDTA, 0.5% Triton X-100, and 1 × protease inhibitors, Roche). After centrifugation at 20,000× *g* for 10 min, 30 µL supernatant was taken out as the input. The other supernatant was incubated with GFP-Trap magnetic beads for 1–2 h, and then the beads were washed four times with NEB buffer. The proteins were eluted from the beads by boiling them with 2 × SDS sample buffer, and they were analyzed by SDS-PAGE and immunoblotted with anti-YFP (TransGen Biotech, Beijing, China, Cat: N20610, 1:5000 dilution) and anti-MYC (Sigma Aldrich, St. Louis, MO, USA, Cat: M4439, 1:5000 dilution) antibodies.

### 4.10. ChIP-PCR

The seedlings of *pro35S:YFP* and *p35S:BZR1-GFP* were grown in liquid 1/2 MS medium for 14 days. Two grams of seedlings were cross-linked in 1% formaldehyde and then by chromatin isolation. The GFP-Trap agarose beads were added into the sonicated chromatin complex. The beads were washed with a low-level salt buffer (50 mM Tris-HCl at pH 8.0, 2 mM EDTA, 150 mM NaCl, 0.5% Triton X-100), a high-level salt buffer (50 mM Tris-HCl at pH 8.0, 2 mM EDTA, 500 mM NaCl, 0.5% Triton X-100), an LiCl buffer (10 mM Tris-HCl at pH 8.0, 1 mM EDTA, 0.25 M LiCl, 0.5% NP-40, 0.5% deoxycholate) and a TE buffer (10 mM Tris-HCl at pH 8.0, 1 mM EDTA) and eluted with elution buffer (1% SDS, 0.1 M NaHCO3). After reverse cross-linking, the DNA was purified and analyzed by ChIP–qPCR. The primers are listed in Appendix A. The fold enrichment was calculated as the ratio between *pro35S:BZR1-YFP* and *pro35S:YFP,* and then, it was normalized to the *PP2A* (At1G13320) gene, which was used as an internal control. The ChIP experiments were performed with three biological replicates from which the means and SD were calculated.

### 4.11. Transient Gene Expression Assays

The protoplast transient assays were performed following previous studies [28,29]. The protoplasts were extracted from mesophyll cells of healthy 4-week-old Arabidopsis leaves. Aliquots of 5 × 10^4^ protoplasts in 0.2 mL MMG solution (0.4 m mannitol, 15 mm MGCL2, and 4 mm MES, PH 5.7) were transformed with 10 μg plasmid DNA using polyethylene glycol method and incubated overnight. The protoplasts were harvested by centrifugation and resuspended in 200 µL of passive lysis buffer (0.5 m mannitol, 20 mm KCL, 4 mm MES PH 5.7) and incubated for 16 H under light. Firefly and Renilla (as internal standard) luciferase activities were measured using a dual-luciferase reporter kit (Promega, Madison, WI, USA).

### 4.12. Accession Numbers

The sequence data from this article can be found in the GenBank/EMBL data libraries under the following accession numbers: BZR1 (At1G75080), BES1 (At1G19350), EBF1 (AT2G25490), EBF2 (AT5G25350), EIN3 (AT3G20770), EIL1 (AT2G27050), PP2A (At1G13320), EXP8 (At2G40610), SAUR15 (AT4G38850), PRE5 (At3G28857), PIF1 (AT2G20180), PIF3 (AT1G09530) and PIF4 (AT2G43010).

## 5. Conclusions

In summary, we found that the F-box protein EBF1 promotes BZR1 protein degradation by the 26S proteasome pathway, inhibiting the BZR1 target genes expression and BR signaling. Meanwhile, BZR1 associates with the promoter of EBF1 and inhibits EBF1 gene expression, inducing BZR1 and EIN3 protein accumulation. Released BZR1 interacts with EIN3 and PIF4, regulating the target genes expression and plant growth. In summary, our study illuminates a molecular mechanism underlying E3 ubiquitin ligase EBF1 which negatively regulates the BR signaling pathway.

## Figures and Tables

**Figure 1 ijms-23-15889-f001:**
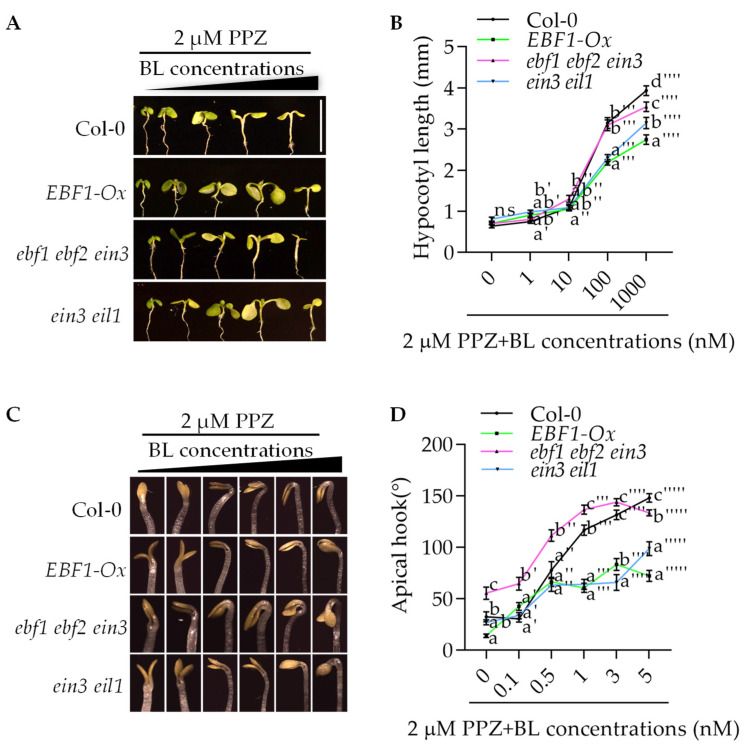
EBF1 inhibited BR-promoted cell elongation and apical hook formation. (**A**,**B**) The hypocotyl length of Col-0, *EBF1-Ox*, *ebf1 ebf2 ein3* and *ein3 eil1* seedlings that were grown on ½ MS medium supplied with 2 µM PPZ plus different concentrations of BL in white light for 7 days. Data are shown as means ± se; *n* = 15. Scale bar = 1 cm. Different letters above bars indicate statistically significant differences between samples (two-way ANOVA followed by post hoc Tukey test, *p* < 0.05). (**C**,**D**) The apical hook curvature of Col-0, *EBF1-Ox*, *ebf1 ebf2 ein3* and *ein3 eil1* seedlings that were grown on ½ MS medium supplied with 2 µM PPZ plus different concentrations of BL in darkness for 4 days. Data are shown as means ± se; *n* = 15. Scale bar = 1 cm. Different letters above bars indicate statistically significant differences between samples (Two-way ANOVA followed by post hoc Tukey test, *p* < 0.05).

**Figure 2 ijms-23-15889-f002:**
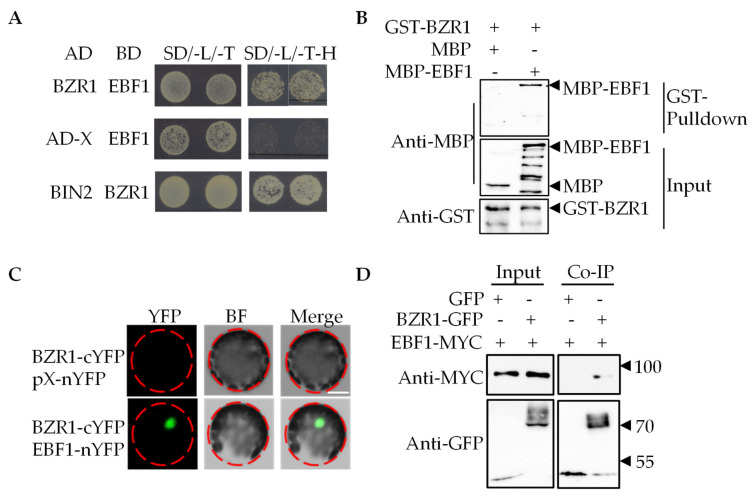
EBF1 interacted with BZR1. (**A**) The yeast two-hybrid assay showed that BZR1 interacted with EBF1 in yeast. (**B**) BZR1 directly interacted with EBF1 in vitro. Glutathione agarose beads loaded with GST-BZR1 were incubated with equal amounts of MBP or MBP-EBF1. Proteins bound to GST-EBF1 were detected by immunoblot analysis with an anti-GST antibody. (**C**) Confocal images of BiFC assays showed that BZR1 interacted with EBF1 in Arabidopsis mesophyll protoplast cells. Scale bar: 20 μm. (**D**) BZR1 interacted with EBF1 in vivo. Immunoprecipitation was performed using GFP-trap beads and Arabidopsis mesophyll protoplasts co-expressing *pro35S:EBF1-MYC* and *pro35S:GFP* or co-expressing *pro35S:EBF1-MYC* and *pro35S:BZR1-GFP*. The immunoblots were probed with anti-MYC or anti-GFP antibodies.

**Figure 3 ijms-23-15889-f003:**
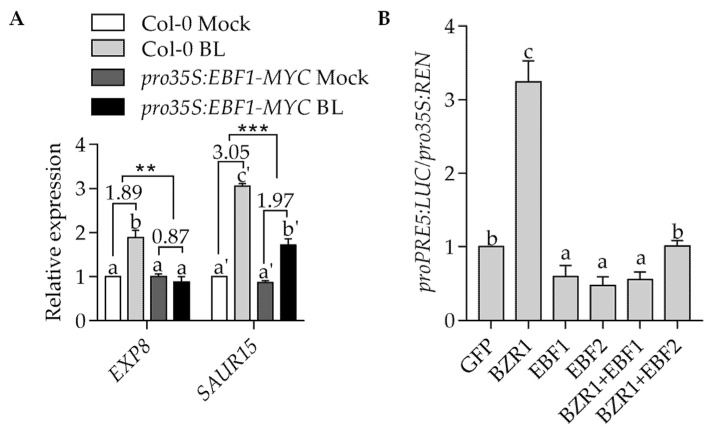
EBF1 inhibited the expression of BZR1 target genes. (**A**) Quantitative RT-PCR showed that EBF1 reduced the BL-promoted gene expression. Total RNAs were extracted from 7-day-old seedlings grown on ½ MS medium with or without 100 nM BL for 3 h. *PP2A* gene was analyzed as an internal control. Error bars are SD of three biologic replicates. Different letters above bars, such as a, b, a’, b’ and c’, indicate statistically significant differences between samples (two-way ANOVA followed by post hoc Tukey test, ** *p* < 0.01, *** *p* < 0.005). (**B**) Transient reporter gene assays showed that EBF1 and EBF2 inhibited the BZR1-promoted expression of *pPRE5:LUC* reporter gene. The dual luciferase reporter construct containing *proPRE5:LUC* (luciferase) and *35S:REN* (Renilla luciferase) was co-transfected with *pro35S:GFP*, *pro35S:BZR1-GFP* and/or *pro35S:EBF1-GFP* and *pro35S:EBF2-GFP* into mesophyll protoplast of wild-type plants. The LUC activity was normalized to REN. Error bars represent standard errors of three biological repeats. Different letters above bars, such as a, b and c, indicate statistically significant differences between samples (one-way ANOVA followed by post hoc Tukey test, *p* < 0.05).

**Figure 4 ijms-23-15889-f004:**
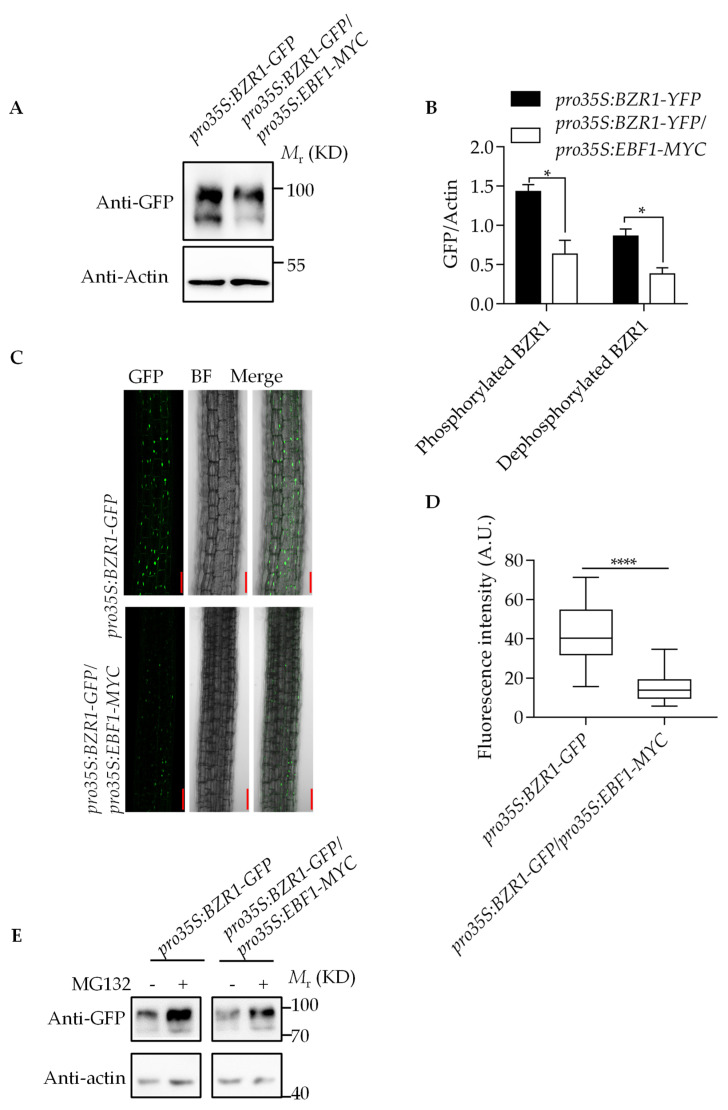
EBF1 promoted the ubiquitination and degradation of BZR1. (**A**,**B**) The protein level of BZR1 in transgenic Arabidopsis plants expressing *pro35S:BZR1-GFP* only or co-expressing *pro35S:EBF1-MYC* and *pro35S:BZR1-GFP*. Seedlings were grown in ½ MS medium for 7 days in in white light. BZR1 was detected with anti-GFP antibody. Actin was used as a control. (**B**) The quantification of BZR1-GFP level displayed in (**A**). GFP/Actin: ratio of immunoblot image intensities between GFP and Actin. Error bars represent the SD of three independent experiments. * *p* < 0.05, as determined by a Student’s *t*-test. (**C**,**D**) Confocal images of BZR1-GFP in the hypocotyl cells. The *pro35S:BZR1-GFP* and co-expressing *pro35S:EBF1-MYC* and *pro35S:BZR1-GFP* transgenic plants were grown on ½ MS medium for 7 days in in white light. GFP signal intensity was analyzed using ImageJ software. Error bars represent the SD. **** *p* < 0.001, as determined by a Student’s *t*-test. Scale bar: 100 μm. (**E**) EBF1 reduced the stability of BZR1 by 26S proteasome pathway. The 10-day-old *pro35S:BZR1-GFP* and *pro35S:EBF1-MYC pro35S:BZR1-GFP* were treated with 50 µM MG132 for 4 h. The BZR1 protein was detected with anti-GFP antibody. Actin was used as a control.

**Figure 5 ijms-23-15889-f005:**
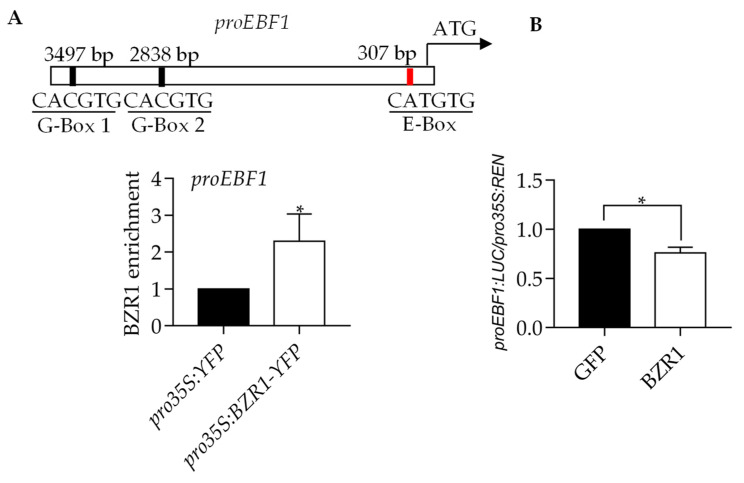
BZR1 regulated the *EBF1* expression level. (**A**) The enrichment of BZR1 to the promoter of EBF1. Schematic illustration of different fragments in the promoter of EBF1. The blank boxes mean the G-box elements (CACGTG) in EBF1 promoter. The red box means the E-box element (CATGTG) in EBF1 promoter. The red box is the fragment selected to do analysis. Seedlings of the *pro35S:BZR1-YFP* and *pro35S:YFP* transgenic plants were grown in the ½ MS liquid medium for 14 days then treated with or without 100 nM BL for 3 hrs. The level of BZR1 binding was calculated as the ratio between *pro35S:BZR1-YFP* and *pro35S:YFP* control, and then normalized to that of the control gene *PP2A*. Error bars represent the SD of three independent experiments. * *p* < 0.05, as determined by a Student’s *t*-test. (**B**) Transient reporter gene assays showed that BZR1 reduced the expression of EBF1. The EBF1 promoter fused to the luciferase reporter gene was co-transfected with *pro35S:GFP* or *pro35S:BZR1-GFP* into mesophyll protoplast of wild-type plants. The LUC activity was normalized to REN. Error bars represent standard errors of three biological repeats. Error bars represent the SE of three independent experiments. * *p* < 0.05, as determined by a Student’s *t*-test.

**Figure 6 ijms-23-15889-f006:**
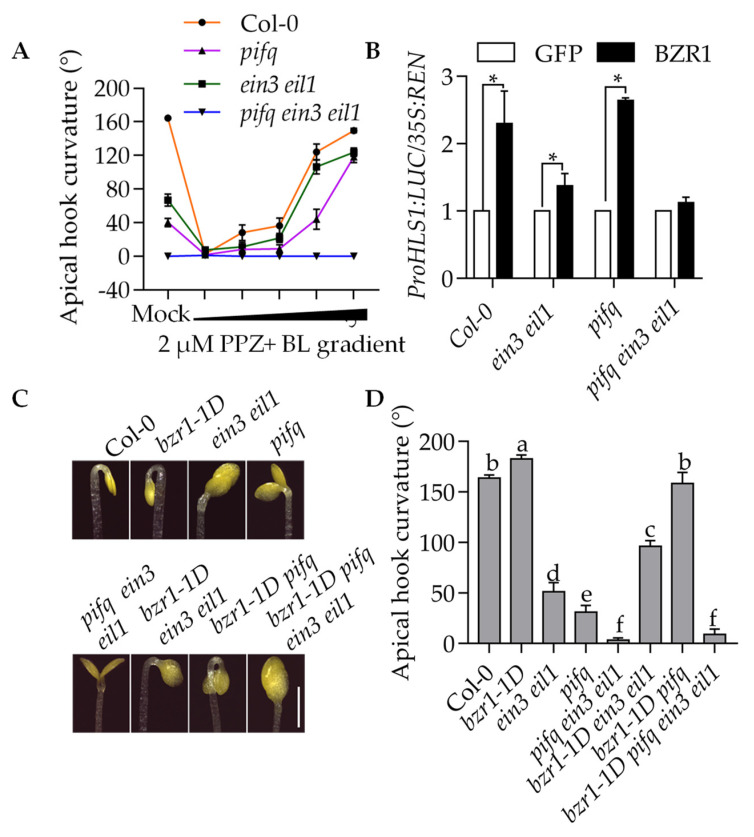
BR and activated BZR1 promotion of hook development require at least one class of transcription factor EIN3/EIL1 or PIFs. (**A**) BR promoted the hook development in *pifq* and *ein3 eil1*, but not in *pifq ein3 eil1*. Seedlings of Col-0, *eni3 eil1*, *pifq* and *pifq ein3 eil1* were grown on the ½ MS medium with or without 2 µM PPZ and different concentrations of BL in the dark for 4 days. The values shown indicated means ± se; *n* ≥ 15. (**B**) Transient reporter gene assays show that the promoting effects of BZR1 on the expression of HLS1 requiring at least one type of transcription factors EIN3/EIL1 or PIFs. The HLS1 promoter fused to the luciferase reporter gene was co-transfected with *pro35S:GFP* and *pro35S:BZR1-GFP* into mesophyll protoplast of wild-type plants, *ein3 eil1*, *pifq*, and *pifq ein3 eil1*. The LUC activity was normalized to REN. Error bars represent standard errors of three biological repeats. * Denotes *p* < 0.05, as determined by a Student’s *t*-test. (**C**,**D**) The gain-of-function mutant *bzr1-1D* suppressed the hook defective phenotype of *pifq* and *ein3 eil1*, but not that of *pifq ein3 eil1*. The seedlings of wild-type plants and indicated mutants were grown on the ½ MS medium in the dark for 4 days. The values shown indicated means ± se; *n* ≥ 15. Different letters above bars indicate statistically significant differences between samples (one-way ANOVA followed by post hoc Tukey test, *p* < 0.05).

## Data Availability

All relevant data are available from the corresponding authors upon request. There are no restrictions on data availability.

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
