# Peer review of "EBF1 Negatively Regulates Brassinosteroid-Induced Apical Hook Development and Cell Elongation through Promoting BZR1 Degradation"

_ijms, 2022, doi:10.3390/ijms232415889_

Round 1

Reviewer 1 Report

In this manuscript, the authors hypothesize that EBF1 promotes the degradation of BZR1 via 26S proteosome pathway through direct interaction. They further study how this interaction modulates the apical hook development and hypocotyl elongation. They finally show how BZR1, EIN3 and PIF4 interact together genetically during the regulation of apical hook development. The data presented supports their hypothesis, and the authors’ findings are novel and significant. However, there are a few issues regarding statistical analysis and manuscript editing which should be resolved before publication.

Major comments:

L 88-89: The authors state that lowered sensitivity of ein3eil1 to BR was partially suppressed by the addition of ebf1 mutation. There are two issues with this statement. First, no statistical analysis was performed for Fig. 1, a one-way ANOVA for a representative BL concentration would be useful, though given the number of plants I’m sure it would support their hypothesis. Second, as far as I can tell the ebf1 mutation does not partially suppress the BL insensitivity. Instead, depending on the dose, it actually makes plants more responsive to BL when it comes to apical hook angle (Fig. 1D). This makes sense, as it involves derepression of BZR1, which overcompensates the ein3eil1 mutation according to authors’ further findings. Nevertheless, a more accurate interpretation of Fig. 1 is needed in the text.

Fig. 3A: The authors state that they have performed two-way ANOVA, however, they show the significance of the columns separately. Because the goal here is to test whether EBF1 overexpression suppresses the induction of EXP8 and SAUR18 in response to BL, the interaction statistics from the two-way ANOVA needs to be used, not Tukey's multiple comparison test.

Manuscript editing:

The authors do not explain many of the abbreviations or the details of their experiments in the text. An example is the inclusion of pifq mutant which was explained only in the methods. Furthermore, the methods lack the description of the Luciferase assays, and the purpose of normalization to Renilla luciferase (which authors simply describe as REN). A casual reader might not be familiar with the terminology. Overall, a better care needs to be taken to 1) Adequately explain abbreviations and experiments in the text and 2) Make sure methods section includes descriptions of all experiments

Finally, there are multiple issues with English. While these are minor, concern the tenses and do not hamper the readability of the text, still another editing of the text for English can be useful. Notable examples include:

1)    Authors switch between homolog and homologue in the text. While both are acceptable, I would choose one and keep it throughout the text.

2)    L29: should be “series” not “a serious”

3)    L34: should be “is exported” not “exports”

4)    L55: should be “is expressed”

5)    L83: should be “In agreement”

6)    L165: should be “were transferred” 

7)    L247: Should be “..by various proteins with…”

8)    L250: Should use “For example” instead of “Such as”

Reviewer 2 Report

The paper is interesting, the experiments are well performed, but the presentation of the results and the use of english must be improved prior to be acceptable for publication. Manuscript needs to be thoroughly revised. Here are some of the typos and mistakes I have found.

Arabidopsis, as the common name of the plant, should not be written in capital, and "Arabidopsis thaliana" as the binomial name, should be written in italics (i.e. lines 49 and 309).

 Line 31: spacing.

Line 54: EBP1 gene "is" expressed.

Line 58: "homolog": check whether they are "paralog" genes.

Line 63: spacing.

Figure 4c: is too small.

Figure 6A: apical.

Line 247-248: Sentence is difficult to understand, please rephrase. Also "various" not "virous".

Line 250: "hormones".

Line 254: "To the promoter".

Line 285: "emerge".

Line 293: "help seedlings".

And many others. 

Reviewer 3 Report

In this study, authors demonstrated that EBF1 promotes BZR1 protein degradation via 26S proteasome pathway by suppressing expression of BZR1-target genes and BR signaling. In addition, authors showed that BZR1 interacting with the promoter of EBF1 and induced protein accumulation of BZR1 and EIN3 while inhibited expression of EBF1 gene expression. In summary, authors showed molecular mechanisms of EBF1 and BZR1 associated with BR associated apical hook development and cell elongation. Overall, the manuscript was nicely written. The purpose of study is fine publication. I have some minor comments as follows.

L31-32 Check the format of sentence.

L49 Arabidopsis thaliana in italics

L63-63 Check the format of sentence.

L100 in vivo and in vitro

L126 and L129 Please use “BZR1-target genes” in the manuscript consistently. (check dash)

L139 EBF1 inhibited expression of BZR1-target genes.

Please indicates a, b, and c in the graph of Figure 3B in the figure legend.

L247 various proteins

Discussion was relatively short. Are there anything to discuss more?

L338 (Transgene, city, country).

L339-341 again for (Thermo, city, country), Biod-rad, Roche

L354 (New England Biolabs, city, country)

L359 Invitrogen

L362 Zeiss  (same mistakes in materials and methods) check them all thoroughly.
